

# Understanding hate speech: the HateInsights dataset and model interpretability

Muhammad Umair Arshad and Waseem Shahzad

Department of Artificial Intelligence and Data Science, National University of Computer and Emerging Sciences, Islamabad, Pakistan

## ABSTRACT

The persistence of hate speech continues to pose an obstacle in the realm of online social media. Despite the continuous evolution of advanced models for identifying hate speech, the critical dimensions of interpretability and explainability have not received proportional scholarly attention. In this article, we introduce the HateInsights dataset, a groundbreaking benchmark in the field of hate speech datasets, encompassing diverse aspects of this widespread issue. Within our dataset, each individual post undergoes thorough annotation from dual perspectives: firstly, conforming to the established 3-class classification paradigm that includes hate speech, offensive language, and normal discourse; secondly, incorporating rationales that outline specific segments of a post supporting the assigned label (categorized as hate speech, offensive language, or normal discourse). Our exploration yields a significant finding by harnessing cutting-edge state-of-the-art models: even models demonstrating exceptional proficiency in classification tasks yield suboptimal outcomes in crucial explainability metrics, such as model plausibility and faithfulness. Furthermore, our analysis underscores a promising revelation concerning models trained using human-annotated rationales. To facilitate scholarly progress in this realm, we have made both our dataset and codebase accessible to fellow researchers. This initiative aims to encourage collaborative involvement and inspire the advancement of the hate speech detection approach characterized by increased transparency, clarity, and fairness.

# INTRODUCTION

In the contemporary digital era, characterized by swift information exchange and inter-connectedness, we are facing new challenges everyday. The dissemination of hate speech and the detrimental consequences it brings is a significant challenge in this regard. Hate

Corresponding author
Muhammad Umair Arshad, umair.arshad@nu.edu.pk

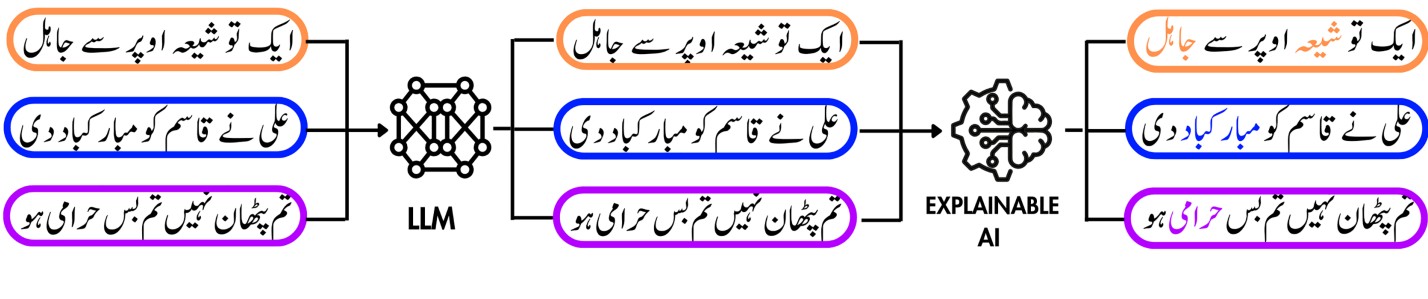

Hate  Neutral  Offensive

**Figure 1** **A visual representation of the process that uses a large language model to classify a tweet as hate/offensive or neutral, followed by the application of an XAI model to highlight the reasoning behind the classification decisions.** Icon source credit: Canva.

[1] Disclaimer: The article contains material that many will find offensive or hateful; however this cannot be avoided owing to the nature of the work.

speech[1]?" [] includes things that encourage violence, discrimination, or unfriendliness towards people or groups based on characteristics such as their race, nationality, gender, religion, or sexual orientation. Such speech has the potential to seriously damage our societal cohesion and the quality of online discourse (*Ali et al., 2022*) invested in addressing this concern, including researchers, policymakers, and technology companies, have been exploring technological solutions, particularly machine learning and artificial intelligence-based solutions, to identify and preemptively combat hate speech. This has led to a growing interest in hate speech detection to place this problem under check. Over the past few years, the community has generated a multitude of datasets (*Ousidhoum et al., 2019*; *Qian et al., 2019*; *De Gibert et al., 2018*; *Bosco et al., 2018*), models (*Zhang, Robinson & Tepper, 2018*), and shared challenges (*Basile et al., 2019*; *Bosco et al., 2018*). These initiatives seek to propel the advancement and development of automated hate speech detection systems.

This article aims to create an automated approach for detecting expressions of hatred in Urdu. The general problem diagram is shown in Fig. 1 (*Ali et al., 2022*). Urdu holds the status of Pakistan's national language, spoken by approximately 300 million individuals globally, despite being a low-resource language (*Bashir et al., 2023*). It reaches far beyond Pakistan, extending to countries like India and Afghanistan, where it is widely spoken and comprehended. The most dominant writing style for Urdu is Nastaliq, which is characterized by a right-to-left orientation and includes diacritics. The primary obstacles inherent in the processing of the Urdu language are the absence of capitalization, a free word-order structure, and its independence from syntactic constituents to impart grammatical meaning. One of the most pressing issues in this context is the utilization of terms such as "black", "Niazi" and "Pathan", which are frequently used to indicate particular races or castes. However, these terms also carry an adverse connotation and can, therefore be utilized to communicate hateful sentiments. Similarly, terms like "non-believer", "Sunni" and "Shia" are commonly used in Urdu and prevalent in hateful remarks as well (*Arshad et al., 2019*). Another challenge lies in the difficulty of obtaining appropriate tokens since Urdu has no well-established tokenizer with the ability to handle

| Urdu Sentence | English Sentence | Label |
|---|---|---|
| ایک تو شیعہ اوپر سے جاہل | One thing a shia and dumb | Hate |
| علی نے قاسم کو مبارکباد دی | Ali congratulated Qasim | Neutral |
| تم پٹھان نہیں تم بس حرامی ہو | You are not a pathan you are a bastard | Offensive |

**Figure 2 Example of hateful, offensive and neutral tweets along with its English translations.**

compound words. Furthermore, the tools available for data pre-processing and cleaning are not suited to languages like Urdu, leading to a persistent challenge in this domain.

The examples of hate speech in Urdu, along with their corresponding English translations are presented in Fig. 2. These examples encompass statements targeted at specific individuals or entities, as well as those that generalize towards entire groups. These instances underscore the gravity of hurtful language in Urdu. It is worth noting that such content can have serious consequences, including hate crimes, irrespective of languages, locations, or ethnicity (*Karim et al., 2020*). Automatized detection, and raising public awareness on hate speech present a multitude of challenges (*Karim et al., 2020*). However, manually examining and confirming a vast amount of online content requires a lot of human effort, in addition to it being a time-intensive task (*Izsak, 2015*).

Another challenge posed by current methods is the lack of clarity regarding the conclusions they reach. As hate speech detection models become more intricate, explaining their decisions is becoming increasingly difficult and equally important (*Goodfellow, Bengio & Courville, 2016*). Legal frameworks like the General Data Protection Regulation (GDPR) in Europe have recently introduced a "right to explanation". This shift demands a move from models that focus solely on performance to models that are interpretable.

In our study, we address the issue of model explainability by simultaneously learning the intended classification and rationales behind human decisions. This approach aims to enhance both aspects in a mutually beneficial manner. As such, judging hate speech should not rely solely on specific words but should consider the context in which these words are used. Even if a text does not contain words generally considered offensive, it can still constitute hate speech. Conversely, the presence of a particular word does not always imply hatred. However, the mere presence of such words can prompt a model to form biased assessments regarding hate speech. This can result in reinforced discrimination against a specific group mentioned in the text (*Sap et al., 2019*). In light of this, it is crucial to eliminate the model's bias towards specific words. Expressions that could lead to biased judgments must be understood in context. This signifies the importance of enabling hate

**Table 1 Contrasting our hate speech dataset with others.**

| Dataset | Labels | Total size sentences | Language | Target labels | Rationals |
|---|---|---|---|---|---|
| RUHSOLD | Abusive/offensive, sexism, religious hate, profane, normal | 10,012 | Roman Urdu | ✓ | ✗ |
| UHATED | Hate speech, offensive, neutral | 15,742 | Urdu | ✓ | ✗ |
| HAOSC | Abusive, non-abusive, threatening, non-threatning | 8,400 | Urdu | ✓ | ✗ |
| HSRU 20 | Hate speech, offensive, neutral | 5,000 | Roman Urdu | ✓ | ✗ |
| **HateInsights** | Hate, offensive, neutral | 11,782 | Urdu | ✓ | ✓ |

speech detection models to make context-based judgments, akin to human discernment. Therefore, these models should be explainable to humans so that the reasoning behind their decisions can be understood.

We have created an extensive dataset named 'HateInsights', encompassing various facets of hate speech. Social media posts from both Twitter and Facebook were collected, and the assistance of students from FAST National University of Computer and Emerging Sciences was enlisted in the process. The annotation covers two dimensions. First, each post is classified into hate, offensive, or normal speech. Second, our annotators were assigned the responsibility of pinpointing text segments that provided support for their classification judgment.

The concept of justification, presented in this context as 'human attention', encompasses a wide spectrum of possible manifestations (*Lipton, 2018*). In this research article, we place specific emphasis on the utilization of rationales, which are textual excerpts from the source text that provide support for a particular categorization. Rationales of this nature have found application in various domains, including common sense explanations (*Fatema Rajani et al., 2019*), e-SNLI (*Camburu et al., 2018*), and numerous other tasks (*DeYoung et al., 2020*). If these rationales do indeed function as convincing justifications for decisions, then models trained to prioritize them could better replicate human-like decision-making processes. For illustrative purposes, refer to Table 1, which showcases examples featuring tokens ('rationales') selected by human annotators, representing their perceived importance for classification.

Our article offers five substantial contributions:

1) Our article involved the creation of the gold standard hate speech detection dataset with Rationals for the Urdu language as it stands.

2) We have established a valuable array of computational resources, encompassing a large annotated dataset, neural language models, open source code, and interpretability techniques, which enable others to advance the field of natural language processing (NLP) research in the Urdu language.

3) We have brought about a compelling advancement in both local-level and global-level explainability while advocating for algorithm based transparency in black-box language models by addressing their opaqueness-which is the inherent problem with them.

4) Our contribution can be succinctly summarized by the induction of 'HateInsights', an innovative benchmark dataset focused on hate speech. This dataset is enriched with both word-level and comment-level span annotations that elucidate the underlying human rationale behind labeling decisions.

## RELATED WORK

### Hate speech

The process of filling the gap between humans and computer language is accomplished through NLP. Significant importance has been given to the global language English in this regard and it is quite justifiable. But on the other hand, regional languages like Urdu, German, Chinese, Hindi and Bengali have not seen their fair share of progress. One of the major tasks of NLP is detection of hate speech on social media platforms targeting minorities and various other groups of cast, color, creed and religion. The dataset utilized in the research pertaining to hate speech predominantly originates from Twitter, primarily due to the linguistic characteristics inherent to the platform. Different datasets contain different types of tweets depending upon the type of classification that is derived from the data. This dataset was developed for classifying the white supremacist hate speech (*Alatawi, Alhothali & Moria, 2021*). The dataset exclusively comprised supremacist tweets, and the research employed two distinct models for hate speech detection. The initial model utilized was Bi-LSTM, yielding notable F1-scores of 0.75. Subsequently, the implementation of BERT resulted in a substantial increase in performance, with the F1-score surging to 0.80. The HateXplain dataset by *Mathew et al. (2021)* was derived from tweets. This dataset uses three types of labels: Hate speech, Offensive and Neutral. This dataset gives an F1-score of 0.687. The speciality of this dataset is that it uses the concept of rationales. The concept of rationales was first given by *Zannettou et al. (2018)*. The RUHSOLD dataset by *Rizwan, Shakeel & Karim (2020)* consists of over 10,000 tweets. It is a Roman Urdu dataset with both coarse grained and fine grained tweets in Urdu language. There are labels of hate speech, and offensive language alongside neutral labels. The concept was that the human annotators marked the part where the text supports their labeling decision. This type of annotation helps in better sentiment analysis. The HASOC dataset curated by *Kumar & Singh (2022)* consists of three languages, namely English, Hindi and Marathi. The dataset is assembled from Twitter and has two tasks. First, hate speech and non offensive labels are offered in all three languages. Then, the dataset for English and Hindi is classified into three labels namely hate speech, offensive and profanity. The F1 measures for Marathi, Hindi and English are 0.91, 0.78 and 0.83, respectively. Another model, proposed in *i Orts (2019)* was trained on dataset of hate speech on Twitter targeting only women and immigrants with more than 10,000 tweets of English and Spanish combined. The model produced up to 0.705 exact match ratio (EMR).

### Hate speech detection in Urdu

The increasing popularity of social media platforms has given rise to a surge in hate speech and racial mockery. To combat such issues, algorithms mainly trained for the English language are usually employed for monitoring and moderation. However, the work on hate

speech detection in the Urdu language remains significantly underdeveloped and necessitates immediate attention. Here is some of the related work, In 2020, *Rizwan, Shakeel & Karim (2020)* introduced a groundbreaking study in Roman Urdu. They developed an innovative model combining BERT and CNN-ngram for detailed classification tasks, achieving a notable F1-score of 0.75. Other researchers have also explored various combinations of feature extraction methods and machine learning techniques to tackle this issue. However, there is still limited research on languages with fewer resources, such as Urdu and Hindi. For instance, *Akhter et al. (2020)* introduced a dataset specifically for detecting offensive language in both Urdu and Roman Urdu, employing character and word-level n-grams to extract features. In 2017, *Mustafa et al. (2017)* and associates analyzed tweets to pinpoint controversial speech, using sequential pattern mining and rule mining to identify frequent patterns and words. Handling of Urdu text encounters multiple obstacles, including its right-to-left orientation (unlike English, which reads from left to right), the absence of capitalization, and its status as a free-word language.

## Explainable hate speech detection

Rationale, as discussed above, is the marking of the text that allows it to be classified as hate speech or offensive text. Using rationales is essential for AI to be able to provide explanations for its results to humans. Some NLP studies indicate that the presence of rationales in text is a key factor in determining the model output. *Hancock et al. (2018)* implemented a generator that generates words, considers rationales, and uses them as input encoders for sentiment classification. The exact words that cause the model to take a decision become highlighted in the output. This is the objective of interpretability-to explain the outputs and internal workings of an AI model in a form that is understandable to humans. This entails including infographics, and feature importance ratings that illustrate how the algorithm reaches its decisions. Detecting hate speech goes beyond merely identifying specific words or word sequences; it requires a certain level of wisdom to accurately determine whether or not something qualifies as hate speech. A method for this is masked rationale prediction (MRP) which predicts human annotations to enhance the reasoning capabilities of AI models (*Kim, Lee & Sohn, 2022*).

# DATASET COLLECTION AND ANNOTATION PROCEDURE

This chapter outlines the essential steps and considerations involved in dataset collection and annotation, which are fundamental to research in fields like machine learning and NLP.

## Dataset curation

To gather data from Twitter based on hashtags and keywords, we utilized the open-source tool Twint (https://github.com/twintproject/twint). For our understanding of common words used to express hate, we conducted a thorough analysis of the words and offensive slurs that were frequently noticed. After gathering these findings, we conducted Twitter searches to gather pertinent tweets. Hence, creating a dataset of around 10,000 tweets. One

noteworthy detail is that the dataset also included tweets in Persian and Arabic since Urdu shares striking similarities with these languages in terms of writing style and common words. As a result, we kept the Urdu tweets and excluded tweets in all other languages. Samples of tweets are shown in Fig. 2.

## Data annotation

For annotation, we utilized students with proficient spoken and comprehension skills in Urdu from both the Master's and Bachelor's programs at FAST National University of Computer and Emerging Sciences Islamabad. The annotation procedure looks into two distinct categories of annotations. Initially, the task involves sorting each annotation into categories: hateful, offensive, or normal. If a consensus among annotators designates a text as hateful or offensive, an additional layer of annotation is deployed to pinpoint the exact words and phrases responsible for the given classification. This analysis provides a clearer understanding of how expressions of hate and offensiveness are demonstrated.

## Annotation instructions

At the beginning of the study, the students agreed to observe and listen to explicit racist, offensive, and hate comments for annotations. Each student received a set of instructions that included guidelines for annotations, the research objectives, and a detailed explanation of each category along with relevant examples. The clear definition provided to each annotator related to the difference between hate speech and offensive language is: "Hate speech attacks certain groups based on characteristics like race or religion, often encouraging hostility. Aggressive language uses strong or harsh words that can scare people, but it doesn't specifically attack any group or lead to widespread unfairness".

Some examples are:

**Example 1: Hate Speech**

**Comment:** "Afghani's should not be allowed to hold office."

**Label:** Hate Speech

**Justification:** The comment targets individuals based on their religious beliefs, suggesting they be excluded from public roles, which promotes discrimination and hostility against a specific religious group.

**Example 2: Offensive Language**

**Comment:** "Shut up, you're an idiot!"

**Label:** Offensive Language

**Justification:** This comment uses direct insults ("idiot") and a command to silence someone ("shut up"), which are aggressive and confrontational but do not target a person based on group identity traits like race, religion, or gender.

## Dataset creation

In this section, we outline our annotation methods, describe how we selected the dataset, and present statistics about the collected data. The detailed statistics of *corpus* is shown in Table 2.

**Table 2** Detailed *corpus* statistics

**Whole *corpus* statistics**

| | |
|---|---|
| No. of unique labelled classes | 3 |
| Total sentences | 11,781 |
| Total words | 265,116 |
| Unique words | 23,934 |
| Maximum length of sentences | 279 |
| Minimum length of sentences | 2 |
| Average length of sentences | 108 Approx |

**Offensive class statistics**

| | |
|---|---|
| Total sentences for offensive class | 5,158 |
| Total words for offensive class | 110,811 |
| Unique words for offensive class | 13,952 |
| Maximum length of sentences for offensive class | 279 |
| Minimum length of sentences for offensive class | 3 |
| Average length of sentences for offensive class | 103 Approx |

**Hate class statistics**

| | |
|---|---|
| Total sentences for hate class | 4,208 |
| Total words for hate class | 112,008 |
| Unique words for hate class | 13,049 |
| Maximum length of sentences for hate class | 274 |
| Minimum length of sentences for hate class | 4 |
| Average length of sentences for hate class | 128 Approx |

**Neutral class statistics**

| | |
|---|---|
| Total sentences for neutral class | 2,415 |
| Total words for neutral class | 42,297 |
| Unique words for neutral class | 8,266 |
| Maximum length of sentences for neutral class | 272 |
| Minimum length of sentences for neutral class | 2 |
| Average length of sentences for neutral class | 84 Approx |

### Main annotation

First of all, we identified potential candidates for the annotation process based on the quality and accuracy of each annotation. To do so, we carried out a testing phase in which every student received a set of text excerpts to annotate and categorize as either hateful, offensive, or neutral. Following task completion, four annotators were chosen from an initial pool of six candidates. The primary annotation process focused on batch tasks, with each round involving 200 text excerpts. These excerpts were distributed among groups of three annotators, and the final label was determined through majority voting among them.

### Class labels

During this phase, we encountered 350 instances where all three annotators selected different categories, rendering majority voting unfeasible, because they are confused in the sentence. So, as a result, these cases were excluded from consideration. For example, in a
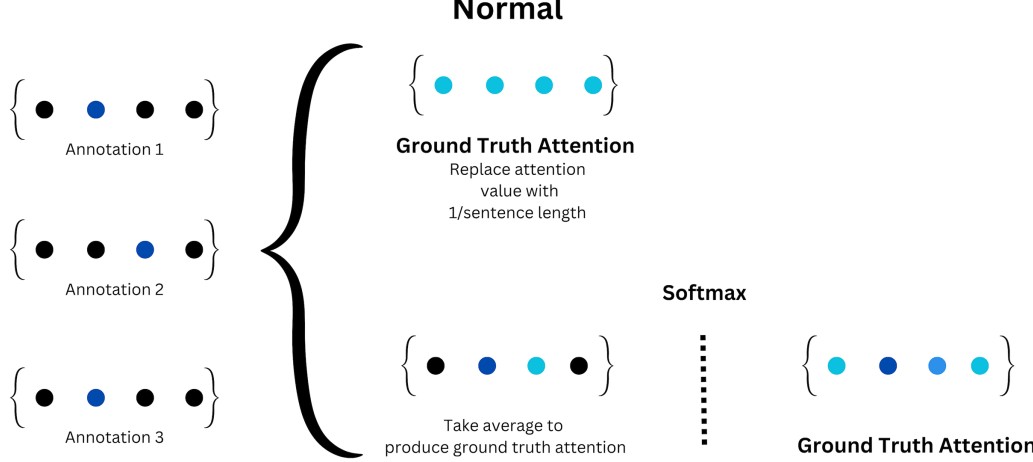

**Figure 3 Ground truth attention process.**

post labeled as "hateful," justifications included the use of derogatory language targeting a specific ethnic group and explicit threats of violence. The instances that received labels of "offensive" or "hateful" were assigned to annotators to come up with two to three explanations justifying the chosen label. Our findings indicated that the average tokens for each sample were seven for hateful speech whereas the average token per post in the whole dataset was found to be 35.8.

### Ground truth attention

This method draws inspiration from *Mathew et al. (2021)* in which each data sample was transformed into an attention vector, essentially a Boolean vector whose length matched the number of tokens in the respective sentence. The tokens in each data sample are indicated as one in the attention vector. This collective attention vector was then normalized using the soft-max function, yielding the ground truth attention. In cases where the label corresponded to the 'normal' category, the attention vector was disregarded, and each corresponding element in the ground truth attention was substituted with '1' to signify a uniform distribution. On the other hand, if the label corresponds to the "hate" or "offensive" categories, the ground truth attention is averaged and a soft-max function is applied to compute the final ground truth attention. Attention vectors generated by attention-based models typically have their sum of elements equal to one. It is very rare for all annotators to choose completely different spans with no overlap. In such cases, we ignore those sentences. When there are overlapping spans, we apply softmax to determine the final span. We illustrate this computation in Fig. 3.

## PROPOSED APPROACH

This section outlines the comprehensive methodology in the proposed work, which includes Data preprocessing and Fine-tuning LLMs. The detailed proposed approach is discussed in the Fig. 4.

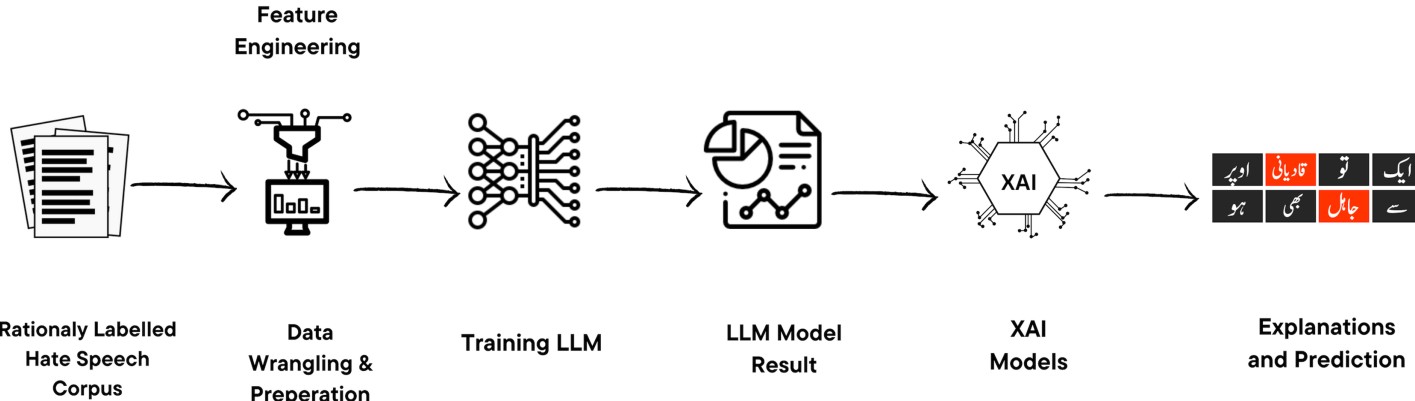

**Figure 4** **Proposed approach for XAI-based classification of hate speech tweets.** Our approach involves training a range of classification models using textual features. We then apply different feature attribution techniques to provide explanations for the predictions made by the model. Icon source credit: Canva.

## Data preprocessing

Following the dataset collection process, the subsequent phase involves data preprocessing and cleaning. We developed and employed a text-cleaning pipeline, designed to terminate unrelated and obstruction-prone data from each tweet.

Each text passed through a specific pipeline which is as follows:

1) The text is parsed and formatted in accordance with the UTF-8 encoding standard, ensuring consistency.
2) Each word in the text is converted to lowercase.
3) Ambiguous characters such as #, +, *, =, HTTP, and HTTPS are removed from the text.
4) Text-based terms are transformed into complete words; for instance, "can't" is changed to "cannot," and so forth.

## Models

We have utilized three advanced and highly sophisticated models to assess the dataset in our study. We have employed these models for classification tasks at two different levels. Initially, these models were tasked with classifying entire sentences into three different classes. Then we integrated them with three explainability models, namely Shap, Lime, and IG, to yield word-level classification results for the hate speech dataset.

### XLM roberta base

The XLM Roberta, designed by Alexis Conneau is a transformer-based masked language model. It has been trained on an extensive *corpus* of multilingual data, totaling approximately 2.5 terabytes and encompassing 100 languages which makes it a formidable choice for text classification tasks (*Conneau et al., 2020*). We fine-tuned this model specifically for hate speech detection on our Urdu dataset. To fine-tune, we replaced the

pre-trained classification head with a new layer of randomly initialized weights which were adjusted during fine-tuning.

### FinBERT

FinBERT is a BERT-based language model which has been fine tuned for hate speech detection. FinBERT features an additional dense layer after the last hidden state for sentiment analysis which makes it perfect for any classification based task (*Araci, 1908*). This model has been configured to classify data into three significant classes, which aligns seamlessly with our own dataset.

### BERT-Hinglish

BERT-Hinglish is another BERT based model, fine tuned on Hinglish dataset to cater specifically to sentiment analysis tasks. It was chosen because of the similarity of the language that it was trained on in comparison to our target language.

### Fine tuning

The various models are evaluated using a consistent data split ratio of 8:1:1 for the training, development, and test-sets. To ensure an even distribution of classes, a stratified split is employed on the dataset. Evaluation results are exclusively reported on the test set, while the development set serves the purpose of fine-tuning hyper-parameters. To fine tune all of these models we leveraged hugging face training APIs. We found that a learning rate of approximately 5e-5, coupled with training batch sizes of 16 and evaluation batch sizes of eight for 5–10 epochs were optimal for training.

## INTERPRETING HATE SPEECH CLASSIFICATION MODELS

The demand for models that not only make accurate predictions but also offer explanations for their decisions is increasing. This trend has resulted in increased research in the field of eXplainable AI (XAI). XAI aims to develop methods that enhance our understanding of how complex models work internally. Basic purpose of XAI is to explain to us why a certain decision was made. This article focuses on applying XAI techniques to the task of classifying Hate/Offensive tweets, using feature attribution methods like Shapley Additive Explanations (SHAP), Local Interpretable Model-Agnostic Explanations (LIME), and other Integrated Gradients (IGs) to provide intuitions and detailed review for the performance of the model in the specific context of the text that is provided.

### Feature attribution methods

The below-mentioned highlighted attribution strategies are types of techniques that decrypt the operational behavior of AI. This specific task is accomplished by assigning some score to the features that are used by these models when making predictions about something. This score assigning process offers valuable insights into how each individual feature contributes to the final prediction of the model. Therefore, this facilitates a deeper understanding of the complex inner mechanisms of these complicated models and allows them to identify potential words that cause biases and inaccuracies. These techniques are classified into two vast categories:

1) Local attribution methods

2) Global attribution methods

Local methods work on illuminating the etiquette of a single prediction, providing in depth review of individual model prediction. Whereas, global methods take a broader view by aiming to enlighten the complete behavior of the model across the entire dataset, offering a complete review on its performance and reasoning. It is quite noticeable that copious feature attribution methods come with their own distinctive strengths and weaknesses. The application has a huge influence on the type of model deployed along with the type of the model. The choice should be made keeping in view the demands of the model. This also includes the elaborateness of the given scenario which ensures that the selected attribution model aligns perfectly with the goals of the model and the objectives of the analysis.

## SHapley additive explanations

SHAP (*García & Aznarte, 2020*) is used in an approach that attempts to represent underlying features of a model without relying completely on its ability of accurate prediction to feature ascription within AI models. Inspecting the SHAP model in depth, we find that it allocates a score to each individual feature based on its contribution to the final prediction. In these types of model predictions, each feature has a role in the final output. The role of SHAP is to determine the level of importance of each feature. Another thing important to notice is that these values are not completely accurate due to the large number of features. Here an approximation is applied and this is where the kernelSHAP algorithm is applied, which is a linear regression-based method that assigns the SHapley values. To compute this value, the role of the feature is determined first, which involves calculating the prediction for all conceivable subsets of features—a task especially for the model. Therefore, an assessment approach is embraced, in which features are taken out by calculating their values from a dataset and in this way resulting in more accurate assignment. Approximation by this method allows us to maintain traceable and efficient assignment of features in AI models.

## Local interpretable model-agnostic explanations

LIME (*Mishra, Sturm & Dixon, 2017*) is a method for interpreting the output of complex and opaque models by creating a simplified prediction through an interpretable model in the context of a particular input. LIME achieves this by constructing an artificially generated dataset and assigning scores to instances that have a match in the original input. Training is done on a linear model using this data to determine how the model works internally. The scores of each feature are then used to determine the role of each feature in the model's prediction for each output. The choice of the model, ranging from a linear model to decision trees, depends entirely on the type of application. It also majorly depends on the complexity of the model's internal working being explained. The model generated by LIME may be based on random sounds or dispersed sounds. In short, the job

of LIME is to interpret a model's prediction without the need to access its training data and underlying architectural structure.

### Integrated gradients

The Integrated Gradients feature attribution method (*Lundstrom, Huang & Razaviyayn, 2022*) is an effective and interpretable feature attribution method for deep neural networks, especially when differentiable models are involved. This method is based on linear interpolation, which requires gradients only in relation to the input and generates a sequence of images from baseline to target image. Gradients for all generated images are then multiplied by an integral of the gradient to compute the scores that are later assigned to the features. This method is highly accordant with the Sensitivity and Implementation Independency Axioms, as the Sensitivity axioms require that if a feature is removed and the overall prediction score changes then the score of that feature is non-zero. The Implementation Independency Hypothesis implies that if two implemented models are functionally equivalent, then the feature attribution scores for them both will be equal regardless of their implementation details.

## EVALUATION METRICS

Determining appropriate evaluation metrics is a crucial process step in assessing the effectiveness of machine learning models and distinguishing between high-performing and subpar models. It is essential to employ a diverse range of evaluation metrics to correctly identify the areas where our models are lacking, thus, enabling us to improve our models further. To achieve this goal, our study incorporates a comprehensive set of evaluation metrics. This article starts off its evaluation by scrutinizing sentence-level classification performance, followed by a more in-depth exploration of word-level classification by the models.

### Sentence-level classification

For sentence-level classification, there were three distinct classes: Neutral, Hate, and Offensive. To gauge the model's performance in sentence-level sentiment analysis, well-established measures of precision, recall, accuracy, F1-score (macro), and AUROC (Area Under the Receiver Operating Characteristic curve) were used.

Precision signifies a model's capacity to accurately identify positive labels, quantifying the proportion of predicted positive outcomes out of the total actual positive labels. On the other hand, recall takes the ratio between correctly predicted positive labels, over the sum of correct and incorrect predicted positive labels. Recall is defined as: $Recall = (TruePositives) \backslash (TruePositives + FalseNegatives)$. The F1-score takes the harmonic mean of these two metrics into a single measure of performance. All metrics are macro-averaged to ensure that each class is given equal weight, which addresses the issue of imbalanced datasets. Additionally, AUROC score is employed to gauge our models' ability to discriminate between different classes.

$$F1\text{-}Score = 2 * \frac{(Precision * Recall)}{(Precision + Recall)}.$$

Here, 'True Positives' denote correctly classified true labels. 'False Negatives' represent true labels incorrectly predicted as false, while 'False Positives' represent false labels incorrectly predicted as true.

## Explainability based metrics

For word-level classification, three distinct explainability models on top of our models were incorporated. In evaluating these models' outcomes, we employed two distinct methodologies. The initial approach involved a one-to-one comparison between predicted and actual labels to calculate accuracy, precision, recall, and F1-scores. The subsequent process was inspired by the ERASER benchmark by *DeYoung et al. (2020)*, employed to compute the explainability of our model. This is assessed using plausibility, which relates to how persuasive or convincing the interpretation is to human observers. Calculating plausibility involves evaluation using two metrics: IOU (Intersection Over Union) F1-score and token F1-score. IOU, adopted from image segmentation calculates the overlap between the predicted bounding box and the ground truth box. In our adaptation, we made sets of consecutive words or a span which had the same label. These sets were then compared with human-annotated labels, categorizing them as true positives (TP), true negatives (TN), false positives (FP), or false negatives (FN) based on a predetermined threshold overlap between labels (set at 0.5 in our study). We then used these TP, TN, FP, FN values to calculate our precision, recall, and F1-scores.

$$IOU = \frac{Intersection\ of\ Predicted\ Labels}{Union\ of\ Predicted\ Labels}.$$

Another measure for explainability is faithfulness.

**Faithfulness:** To evaluate faithfulness, we employ two metrics: comprehensiveness and sufficiency (*DeYoung et al., 2020*).

**Comprehensiveness:** In order to assess the comprehensiveness of our approach, we undertake the following procedure. For each post $x_i$, we generate a contrasting example $\tilde{x}_i$ by eliminating the predicted rationales $r_i$ from $x_i$. Let $m(x_i)_j$ denote the initial prediction probability assigned by the model $m$ for the class $j$.

We then establish $m(x_i \backslash r_i)_j$ as the predicted probability for $\tilde{x}_i = (x_i \backslash r_i)$ (which is $x_i$ with the rationales removed) by the model $m$ for the class $j$. It is expected that the model's prediction would decrease after removing the rationales. This can be quantified as follows:

$$Comprehensiveness = m(x_i)_j - m(x_i \backslash r_i)_j.$$

A higher comprehensiveness score suggests that the rationales had a substantial impact on the prediction.

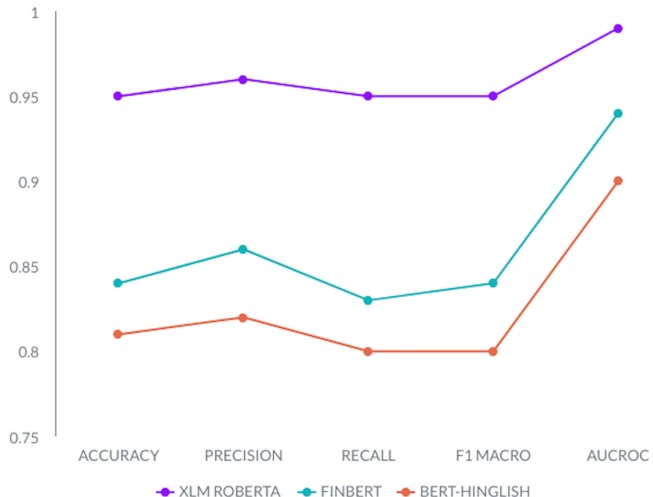

**Figure 5** Accuracy, precision, recall, F1-score, and AUROC comparison of all classification models, i.e., XLM-Roberta, FinBert, and Hinglish.

**Sufficiency**. This metric quantifies the extent to which the extracted rationales provide enough information for a model to make a prediction. We can express this measurement as follows:

$$Sufficiency = m(x_i)_j - m(r_i)_j.$$

In this equation, a higher sufficiency score indicates that the rationales sufficiently support the model's prediction.

## RESULTS AND DISCUSSION

Interpreting the decision-making process of large language models is a crucial aspect of analysis, aiming to uncover how these classifiers arrive at their predictions. In this section, we present the outcomes of our feature attribution analysis for a group of Large Language Models (LLM's) employed in hate speech classification. We delve into the detailed interpretations of SHAP, LIME, and IG feature attribution values, offering valuable insights into how each model leverages input features to make predictions. Furthermore, we investigate the typical prediction paths followed by these classifiers and their implications for the overall decision-making process.

### Classification

The Fig. 5 results comparing three models—XLM-Roberta, Hinglish, and FinBert—across performance metrics such as accuracy, precision, recall, F1-score, and AUROC consistently show that XLM-Roberta outperforms the other models in all aspects. XLM-Roberta achieves accuracy of 0.95, precision of 0.96, recall of 0.95, F1-score of 0.95, and AUROC scores of 0.99, indicating its superior performance in identifying hate speech content. These exceptional results firmly establish XLM-Roberta as the top-performing model among the three for the critical task of hate speech detection.

| English Translation | Urdu Sentence | Predicted Label | Actual Label |
|---|---|---|---|
| Brother, happy birthday. May Allah keep you in His protection. | بھائی اپنی سالگرہ مبارک اللہ اپنی حفاظت میں رکھے | Neutral | Neutral |
| Black apostate, Shia infidel | کالا مرتد شعیہ کافر | Hate | Offensive |
| Shia infidel, Qadiani liar, what's your problem? | شعیہ کافر قادیانی جھوٹے کیا مسئلہ ہے اپ کا | Offensive | Hate |

**Figure 6** **Some examples classified by XLM-Roberta.** The first two examples are correctly classified by our model, while the other two examples are wrongly classified by our model.

On the other hand, BERT-Hinglish lags behind in our evaluation. This discrepancy may be attributed to its training on the Romanized version of the Hindi language, while our task focuses on standard Urdu. It's worth noting that XLM-Roberta's multilingual capabilities, trained on approximately 100 languages, likely contribute to its outstanding performance in our study.

Our best performing model, XLM-Roberta-Base, has exhibited remarkable proficiency in differentiating between different classes. The model excels in distinguishing between various classes in most cases. However, it experiences challenges when discriminating between the Hate and Offensive classes, as these categories exhibit substantial overlap and linguistic similarity, as evident in the sentences shown in Fig. 6. It's worth highlighting that even when the model mislabels text, it often exhibits a notable decrease in prediction probability for the predicted class which is indicative of its underlying awareness of distinction between the classes.

The mislabeling for XLM-Roberta-Base, for the Offensive and Hate classes, stands below 3.5%, notably a superior performance in comparison to the alternative models, Bert-Hinglish and Finbert. XLM-Roberta-Base excels in identifying the Neutral class, with misclassifications accounting for a mere 1.61% of the examples. This accuracy is partly attributed to the straightforward meanings of words typically encountered in Neutral sentences. Conversely, the Hate and Offensive classes often involve context dependent words with nuanced meanings which leads to heightened classification complexity. Furthermore, the linguistic overlap between words in the Offensive and Hate classes exacerbates the model's challenge in distinguishing between them. In conclusion, our XLM-Roberta-Base showcases impressive capabilities in multiclass text classification, with misclassification occurrences largely confined to the intricacies of differentiating between closely related classes.

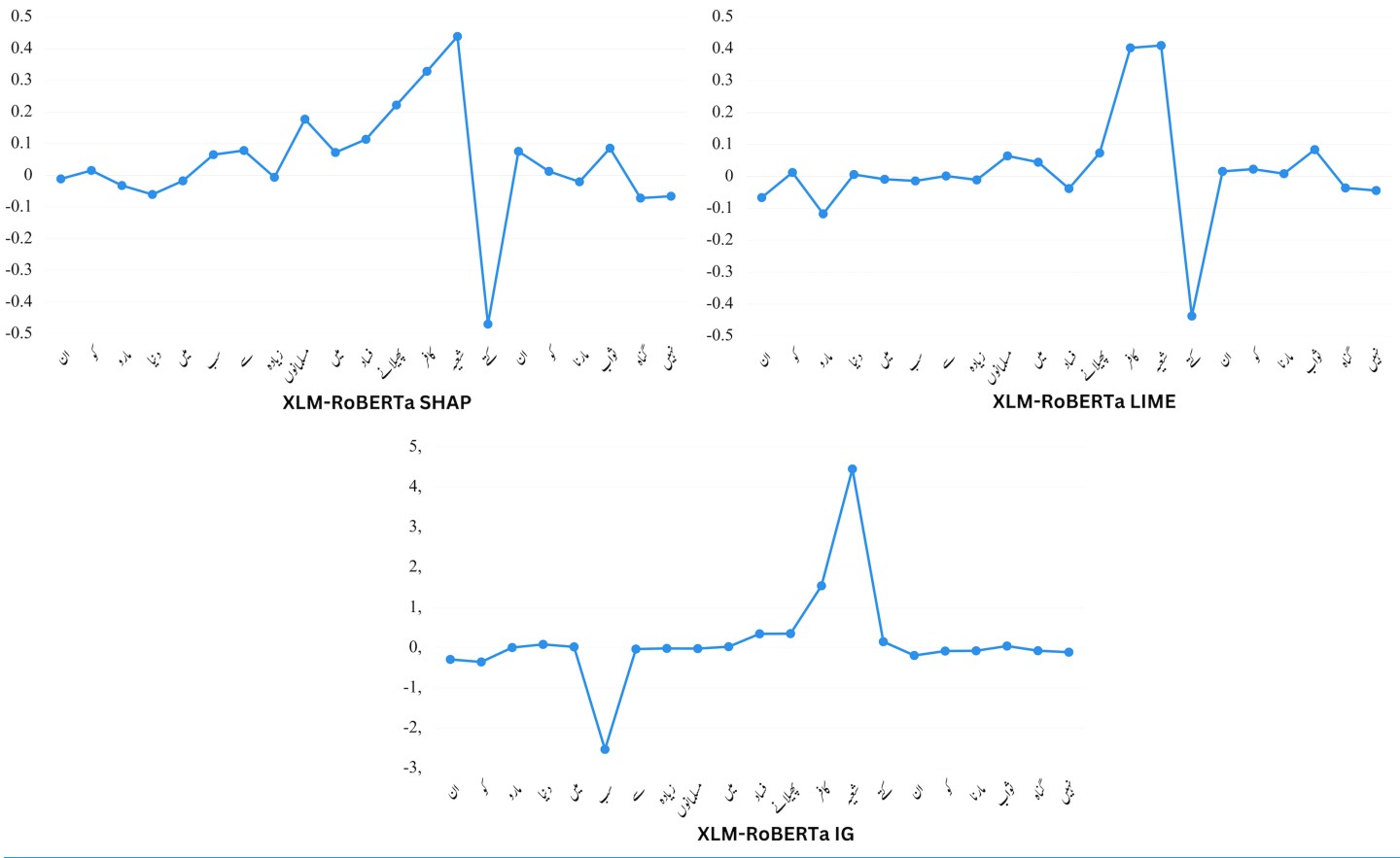

**Figure 7** An illustration of word attribution scores for XLM-RoBERTa with SHAP, LIME, and IG.

## Explainability

Figure 7 presents detection results from models trained with human rationale. The visualized rationales in the form of LIME results serve as the basis for measuring explainability-based scores. For human-ground truth, the highlighted token represents the rationale identified as important by human annotators for prediction.

The top row displays the tokens ("rationales") chosen by human annotators as crucial for classification. The subsequent three rows exhibit the significant tokens (determined using LIME) that aided various models in their classification tasks. Explicit hate speech containing obvious derogatory expressions is relatively straightforward to assess. As demonstrated in the first example in Fig. 8, all models perform admirably. Both human rationale and model rationale tend to emphasize specific abusive words. However, the rationales generated by the proposed XLM-Roberta model align more closely with the ground truth compared to other baseline models.

The detection results of FinBERT or Hinglish appear to be inaccurate for certain sentences. In contrast, Xlm-Roberta demonstrates high accuracy, primarily due to its rationale, which closely resembles human reasoning more than the other models.

| No. | Model | Example sentence and Rationale | Label |
|---|---|---|---|
| 1 | Human | ان کو مارو دنیا میں سب سے زیادہ مسلمانوں میں فساد پھیلانے کافر شعیہ کتے ان کو مار ناثواب گناہ نہیں<br><br>Kill them, the infidel Shia dogs who spread the most corruption among Muslims in the world. Killing them is not a reward for sin | Hate |
| | FinBert | ان کو مارو دنیا میں سب سے زیادہ مسلمانوں میں فساد پھیلانے کافر شعیہ کتے ان کو مار ناثواب گناہ نہیں | |
| | Hinglish | ان کو مارو دنیا میں سب سے زیادہ مسلمانوں میں فساد پھیلانے کافر شعیہ کتے ان کو مار ناثواب گناہ نہیں | |
| | Xlm-Roberta | ان کو مارو دنیا میں سب سے زیادہ مسلمانوں میں فساد پھیلانے کافر شعیہ کتے ان کو مار ناثواب گناہ نہیں | |
| 2 | Human | بہت خوب بات ان بے حیا لبرل انٹیوں ساتھ یہی روایہ رکھنا چاہیے<br><br>It is very good to have the same attitude with these shameless liberal aunties | Hate |
| | FinBert | بہت خوب بات ان بے حیا لبرل انٹیوں ساتھ یہی روایہ رکھنا چاہیے | |
| | Hinglish | بہت خوب بات ان بے حیا لبرل انٹیوں ساتھ یہی روایہ رکھنا چاہیے | |
| | Xlm-Roberta | بہت خوب بات ان بے حیا لبرل انٹیوں ساتھ یہی روایہ رکھنا چاہیے | |
| 3 | Human | کتنے بغیرت بے شرم بے حیا حرام کھا کھا کر تم کو مر نا یاد نہیں<br><br>How shameless and immodest you are, forgetting death while living off the forbidden. | Offensive |
| | FinBert | کتنے بغیرت بے شرم بے حیا حرام کھا کھا کر تم کو مر نا یاد نہیں | |
| | Hinglish | کتنے بغیرت بے شرم بے حیا حرام کھا کھا کر تم کو مر نا یاد نہیں | |
| | Xlm-Roberta | کتنے بغیرت بے شرم بے حیا حرام کھا کھا کر تم کو مر نا یاد نہیں | |

**Figure 8** **An illustration of predicted rationales by various models in contrast to human annotators.** Tokens highlighted in orange signify the ones both the human annotator and the model considered crucial for the prediction, while tokens highlighted in magenta represent tokens that the model deemed significant but were not identified as such by the human annotators.

The detailed results are shown in the Table 3. Our research consistently found the XLM-Roberta-Base model to be exceptional in both sentence and word-level classification tasks. The model consistently delivers strong IOU and comprehensive performance scores. However, the model does exhibit some limitations when it comes to sufficiency scores. In the context of sufficiency, it is worth noting that the Bert-Hinglish base model, when integrated with all available explainability models, exhibits high scores. Nevertheless, its comprehensive and plausibility scores do not rank it as the top choice in our evaluations making it the least favored choice for use in text classification.

For token-level plausibility, our experiments show that XLM-Roberta-Base, when employed with LIME, achieves the highest accuracy. On the other hand, when evaluating token-level plausibility using the F1-score, XLM-Roberta-Base combined with Shap outperforms all other alternative models. Furthermore, our analysis of IOU levels highlights the effectiveness of Information Gain (IG) when integrated with the XLM-Roberta model, as it yields a superior F1-score.

One striking observation is the relatively low recall (IOU) achieved with the LIME explainability model. This phenomenon can be explained by our constrained feature

**Table 3 Results of model performance: we employed SHAP, LIME, and IG methods for token selection in the calculation of explainability.** Model performance metrics primarily refer to classification scores, while explainability metrics encompass plausibility and faithfulness measurements. The bold entries reflect the highest score.

| Model | Performance | | | | | Explainability | | | | | | |
|---|---|---|---|---|---|---|---|---|---|---|---|---|
| | | | | | | Plausibility | | | | | Faithfulness | |
| | Acc. ↑ | Prec. ↑ | Recall ↑ | F1 Macro ↑ | AUROC ↑ | Acc. (Token) ↑ | F1 Macro (Token) ↑ | Prec. (IOU) ↑ | Recall (IOU) ↑ | F1 (IOU) ↑ | Comp ↑ | Suff ↓ |
| Bert-Hinglish (IG) | 0.81 | 0.82 | 0.80 | 0.80 | 0.90 | 0.45 | 0.45 | 0.51 | 0.64 | 0.57 | 0.05 | −0.01 |
| Bert-Hinglish (Lime) | 0.81 | 0.82 | 0.80 | 0.80 | 0.90 | 0.64 | 0.40 | 0.87 | 0.01 | 0.02 | −0.07 | **−0.003** |
| Bert-Hinglish (Shap) | 0.81 | 0.82 | 0.80 | 0.80 | 0.90 | 0.53 | 0.52 | 0.60 | 0.61 | 0.60 | 0.09 | −0.37 |
| FinBERT (IG) | 0.84 | 0.86 | 0.83 | 0.84 | 0.94 | 0.46 | 0.46 | 0.54 | 0.79 | 0.64 | 0.59 | 0.02 |
| FinBERT (Lime) | 0.84 | 0.86 | 0.83 | 0.84 | 0.94 | 0.62 | 0.46 | 0.77 | 0.08 | 0.14 | 0.43 | 0.08 |
| FinBERT (Shap) | 0.84 | 0.86 | 0.83 | 0.84 | 0.94 | 0.47 | 0.46 | 0.51 | 0.49 | 0.50 | 0.69 | 0.02 |
| XLM Roberta (IG) | **0.95** | **0.96** | **0.95** | **0.95** | **0.99** | 0.50 | 0.50 | 0.59 | **0.80** | **0.68** | 0.52 | 0.13 |
| XLM Roberta (Lime) | **0.95** | **0.96** | **0.95** | **0.95** | **0.99** | **0.64** | 0.53 | **0.87** | 0.23 | 0.36 | 0.50 | 0.12 |
| XLM Roberta (Shap) | **0.95** | **0.96** | **0.95** | **0.95** | **0.99** | 0.55 | **0.66** | 0.56 | 0.44 | 0.49 | **0.74** | 0.07 |

selection, a conscious choice made to accommodate limited hardware resources available without compromising the overall model evaluation process.

# CONCLUSION

This article introduces DeepHateExplainer, a novel explainable method designed for hate speech detection in the under-resourced Urdu language. DeepHateExplainer utilizes ensemble prediction techniques and achieves an impressive F1-score of 92%, surpassing several baseline models, including DNNs and LLMs. Additionally, we present a comprehensive analysis of explainable approaches, including LIME, SHAP, and IG, applied to Urdu data. We conduct a comparative study of these three methodologies, assessing their suitability for providing both global and local explanations. To achieve a fundamental comparison, we utilize SHAP's capacity for sample-based and global explanations, leveraging LIME's mean word importance statistics to infer global significance. This approach enables us to conduct a robust evaluation.

A possible improvement of the proposed approach refers to the inclusion of additional features. While the proposed approach only considers the text within the tweet, sentiment information and lexical characteristics (*e.g.*, the usage of uppercase or emoji) have been shown to be important clues for hate-related tasks. Furthermore, to stay at the forefront of advancements in language models, we have incorporated the latest developments, including models like LlaMA, which augment our approach's capabilities in understanding

and identifying hate-related content within tweets. This combination of traditional features and cutting-edge language models reinforces the robustness and effectiveness of our approach.

### Funding
The authors received no funding for this work.

### Competing Interests
The authors declare that they have no competing interests.

### Author Contributions
- Muhammad Umair Arshad performed the experiments, analyzed the data, performed the computation work, prepared figures and/or tables, and approved the final draft.
- Waseem Shahzad conceived and designed the experiments, authored or reviewed drafts of the article, and approved the final draft.

### Data Availability
The data is available at GitHub and Zenodo:

- https://github.com/umairarshad786/XAIHate

- Arshad, M. U. (2024). Hate Speech [Data set]. Zenodo. https://doi.org/10.5281/zenodo.12748652.

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
