# Peer review of "Understanding hate speech: the HateInsights dataset and model interpretability"

_PeerJ Computer Science, doi:10.7717/peerj-cs.2372_

## Round 0.1 · original submission · Major Revisions

The Reviewers agree that the topic of the paper is interesting and worth exploring.

However, a complete revision of the quality of the presentation of its content is required to allow a better delivery and understanding of the proposed study.
Please, carefully consider the Reviewers’ suggestions, especially concerning the presentation of the novelty of your proposal and the preparation of specific research questions to complement the discussion of your contributions.

Concerns have been raised on the experimental design and the validity of the findings and must be addressed.

Please, add the translation of the non-English content for clarity.

Carefully consider the Reviewers' suggestions and comments, providing a point-by-point response to them.

Reviewer 1 ·

Basic reporting

The authors propose the HateInsights dataset, which includes posts in Urdu that are labeled for hate speech, offensive language, and normal discourse. Additionally, it incorporates the rationales for the assigned labels. The authors also propose a classification model enriched by interpretability techniques.
The paper would benefit from a general revision since it results quite difficult to read. The state of the art, is quite confusing and the novelty of the proposed approach does not emerge since it is presented as a composition of existing approaches (e.g. transformer-based models and explainability techniques).
In particular, all the references should be better encapsulated within the text to improve readability (e.g. in line 33 Ali et al. (2022) those invested in.. --> In Ali et al. (2022) the authors … ). As a consequence, the paper is quite difficult to read, especially in the section dedicated to related works. Moreover, there does not seem to be an order in which the works are presented since works with different approaches and datasets are listed alternating different characteristics. Although the works in Urdu are a few (in comparison to English) at least the main ones should be included in the state of the art.

Experimental design

For what concern the dataset section, the inclusion of the adopted guidelines would allow a better evaluation and comprehension of the adopted procedure. Moreover, is not clear how the explanation justifying the chosen labels are included within the dataset; additional information in section 3.4.2 are necessary, possibly also including an example of such explanations.
Section 3.4.3 , although placed in the dataset section, seems to focus on the adopted method. I suggest replace the subsection and enriching it to better clarify how the attention vector is created.
The approach proposed for the evaluation, involving sets of consecutive words, presented at line 351 is not clear.
Finally, al the figures and tables in the paper are correctly referenced and discussed in the paper, but can be better placed within the text to improve readability.

Validity of the findings

The novelty of the proposed approach does not emerge since it is presented as a composition of existing approaches (e.g. transformer-based models and explainability techniques).
I suggest highlighting the novelty and strenght of the proposed approach and improve the section dedicated to results to better contextualize and discuss achieved results.

Additional comments

Some final typos and considerations:
- The open-source tool “Twint” mentioned at line 165 should be referenced, eventually in a footnote.
- In section 4.2, several model al listed, in particular XLM Roberta Base and FinBERT are fine tuned on sentiment analysis. On which dataset? Why sentiment analysis? Wouldn’t it be better to finetune it on hate-related tasks?
- At lines 341 and 342 a definition of false positive and false negatives is provided, I suppose there is a mismatch in the definitions.
- At line 365, the class is called m, instead of j
- In the first point of the list, at line 98, the authors state that the proposed dataset is the largest one, although, from table 1, the UHATED seems to share similar characteristics and a bigger number of samples.

Cite this review as

Reviewer 2 ·

Basic reporting

The paper is interesting but it needs to be improved in terms of its quality and needs to be proof read to fix typos and cite properly references (see below some of the examples but this should be applied to every time a reference is cited). In the references, all arXiv ones should be substituted with their equivelent peer-reviewd ones. Moreover, some recent references on bias analysis in hate speech (to eliminate the model’s bias towards specific words) [1] and XAI [2] should be included in the section on related work.

In the revised version of the paper, the authors should also describe how the text is finally annotated if the span selected by the different annotators is different. The authors could find interesting this related work 34]. Moreover, the quality of the figures should be improved and the translation into English of the examples also added (e.g. Figures 6 and 8).

In line 237 the authors comment that they fined tuned the model for sentiment analysis: why not for hate speech?

In the analysis of the results, it would be interesting if the authors could investigate the presence of implicit hate speech cases and how they have been (mis)classified, being the detection of implicit hate speech much more challenging, e.g. [4].

In the references, arXives should be substituted by their peer-reviwed versions.
In line 502: where was the reference published?


[1] De La Peña G., Rosso P. (2023) Systematic Keyword and Bias Analyses in Hate Speech Detection. In: Information Processing & Management, vol. 60, issue 5
[2] Attanasio G., Pastor E., Di Bonaventura C., and Nozza D. (2023). Ferret: a Framework for Benchmarking Explainers on Transformers. In Proceedings of the 17th Conference of the European Chapter of the Association for Computational Linguistics: System Demonstrations, pages 256–266, Dubrovnik, Croatia. Association for Computational Linguistics. https://aclanthology.org/2023.eacl-demo.29/
[3] Joshi, M., Chen, D., Liu, Y., Weld, D. S., Zettlemoyer, L., & Levy, O. (2020). SpanBERT: Improving Pre-training by Representing and Predicting Spans. Transactions of the Association for Computational Linguistics, 8, 64-77.
[4] Frenda S., Patti V., Rosso P. (2023) Killing me Softly: Creative and Cognitive Aspects of Implicitness in Abusive Language Online. In: Natural Language Engineering (JNLE), 29(6): 1516 - 1537

Experimental design

In this paper the authors introduce the HateInsights dataset where each text has been labelled as hate speech, offensive language, and normal disocurse. Moreover, the authors introduce DeepHateExplainer, a novel explainable method designed for hate speech detection. Special emphasis is given on explainability in order to increase transparency, claarity, and fairness. The authors present a comprehensive analysis of explainable approaches, including LIME, SHAP, and IG in order to model also plausability and faithfulness. What make this work especially valuable is that it focuses on Urdu data, being Urdu still a low-resourced language in NLP especially from the perspective of XAI models.

Specific research questions should be defined at the end of Sect. 1 instead of generic contributions.

Validity of the findings

Some interesting insights were found such as the importance of sentiment analysis and lexical characteristics (e.g. emojis and the usage of uppercase). Several LLMs (XLM-RoBERTa, FinBERT, Hinglish) have been used to carry out the experiments and the best results have been obtained by LLaMA.

Additional comments

The paper should be proof read in order to fix the many typos. Some examples are below.

Line 33: discourse. Ali et al. (2022) Those invested -> Discourse (Ali et al., 2022) invested
38 and 39: (Ousidhoum et al. (2019) … -> (Ousidhoum et al., 2019) etc. (the ssame for the other references9
44: low-resource language. Bashir et al. (2023) It reaches -> low-resource language (Bashir et al., 2023). It reaches
53: as well. Arshat et al. (2019) Another challenge -> as well (Arshat et al., 2019). Another challenge (same in line 126)
61, 62, 64, 89, 122: same way to cite references
Usage of uppercase in Table 1
124: Bil-STM -> Bi-LSTM
151: NLP (Natural Language Processing) -> Natural Language Processing (NLP)
171, 212: missing . at the end
219: category -> categories
258 Explainable AI (XAI) -> eXplainable AI (XAI)
382 The fig. 5 -> Fig. 5
396: in the figure 6 -> in Figure 6
406: In conclusion, On -> In conclusion, on
410: Fig. 8 -> Fig. 7
417: in fig. 8 -> in Fig. 8
424: in the table 3 -> in Table 3

Cite this review as

---

## Round 0.2 · Major Revisions

The Reviewer has raised major concerns about the revised version of the manuscript.

In particular,

- The novelty of the proposed approach is questioned. Please, clearly highlight the contributions of the proposed work and how it distinguishes from other literature works.
- The declaration of reproducibility requires both a clarification of the methodology and the sharing of a link to the used resources.
- Please, provide a clear definition of the concepts highlighted by the reviewer, such as, “hate speech” and “aggressive language”.
- Please, clarify the evaluation procedure following the reviewer’s suggestions.
- Please, provide clear examples. In case of vulgar content, consider substituting it with mild contents. You can also consider warning the reader of possible hateful, offensive, and vulgar text reporting at the beginning of the manuscript (for example it seems that the texts present in Figure 8 are quite offensive and hateful).

The authors need to address all the comments made by the reviewers. Divide the comments in a point-by-point response but ensure that the whole reviewers’ reports are present. This would improve the clarity of your responses and ensure that the reviewers are not left with the doubt of not having their concerns addressed.

In the tracked changes version of the manuscript all the changes need to be clearly highlighted not only as comments. For example, the authors can strikeout the text in case of original text removal, and colour/highlight new or modified text. This would allow a better understanding of the modifications applied to the original version of the manuscript.

Notice that the explanations given just as comments are not provided in the final version of the manuscript, and thus would not be useful to the readers to better understand the topics that are highlighted as not clear by the reviewers.

Please, read all the reviewers’ comments, including the ones from the previous revision round, to include your responses in the manuscript in a clear, formal, and descriptive manner.

Reviewer 1 ·

Basic reporting

I thank the authors for addressing the concerns highlighted in the previous review.

The novelty of the proposed approach is quite limited: the authors finetune existing BERT-based models on the proposed dataset. The paper present the work as a composition of existing approaches (e.g. transformer-based models and explainability techniques).

For wat concerns reproducibility of the work, the authors state “To facilitate scholarly progress in this realm, we have made both our dataset and codebase accessible to fellow researchers.”, however any link to access such material is present.

Experimental design

As I pointed out in my previous review, information about the adopted guidelines would allow a better evaluation and comprehension of the adopted procedure. Within the State of the Art many definition of “hate speech” and “aggressive language” have been proposed, eventually overlapping. I therefore suggest the author to include a clear definition of such concepts. Moreover, it is not clear weather such concepts were left to the interpretation of the single annotators or if examples/definitions were provided in the annotations guidelines.
For what concerns the explanation gathered, I understand the concerns of the author in reporting vulgar content, however an example of such “justifications” would clarify how the explanation are included within the dataset.
The example added only included additional explanations or simple example of annotations excluding the justification labels.

Validity of the findings

The approach proposed for the evaluation, involving sets of consecutive words, as I pointed out in my previous review, is not clear. However, the authors only addressed such concerns in the “answer to the reviewers” document and in some side notes (“Basically, it’s very rare for all annotators to choose completely different spans with no overlapping. So, in the case of overlapping spans, we apply softmax to determine the final span.“), without editing the actual paper, that remain in my opinion unclear, making the whole evaluation and result sections difficult to understand. Moreover, such additional details do not clear up the adopted evaluation procedure that remain quite confusing.

Additional comments

Some small typos are still present, for instance, at line 369 a (?) appears probably indicating a missing citation.
Finally, all the figures and tables in the paper are correctly referenced and discussed in the work, however, their positioning within the text can be improved.

Cite this review as

---

## Round 0.3 · accepted · Accept

I thank the Authors for carefully addressing all the Reviewers’ comments. The work is now ready for publication.

The responses to the first revision round have been carefully checked and the final concerns of the Reviewers addressed.

- The presentation of the work has greatly improved.
- The description of the contribution of your work as well as of the proposed methodology are now clear.
- The concerns on the experimental design and validity have been addressed.
- All the required definitions have been provided.
- The provided examples are satisfactory.

There are only some minor typos (e.g., line 32) that you can easily address with the editorial support team.

Reviewer 1 ·

Basic reporting

I thank the authors for addressing my concerns. The paper is now overall improved.
The introduction of definitions and examples makes the proposed approach clearer and allow for a better contextualization and comparison within the state of the art.
The changes introduced make the proposed approach clearer and highlight innovations in comparison to existing approaches in the same domain.

Experimental design

-

Validity of the findings

-

Additional comments

-

Cite this review as